# Effects of Morphology and Solvent/Temperature on THz Spectra: Take Nucleosides as Example

**DOI:** 10.3390/molecules28041528

**Published:** 2023-02-04

**Authors:** Fang Wang, Haifeng Lin, Jiawen Tong, Junbin Tai, Jiaen Wu, Yaru Yao, Yunfei Liu

**Affiliations:** 1College of Electronic Engineering, Nanjing XiaoZhuang University, Nanjing 211171, China; 2College of Information Science and Technology, Nanjing Forestry University, Nanjing 210037, China

**Keywords:** FTIR, nucleosides, PBC-GEBF, hydrogen bonds, solvation effects

## Abstract

Water molecules were easy to combine with organic molecules and embed into the lattice of solid molecules to form a hydrate. Compared with anhydrous compounds, a hydrate has completely different physical and chemical properties. In this paper, terahertz (THz) spectra of five nucleosides in the solid and liquid phases were studied experimentally by Fourier-transform infrared spectroscopy (FTIR) in the frequency of 0.5–9 THz. In addition, the lattice energy, geometric structure, and vibration spectrum of the molecular crystal of the nucleosides were analyzed theoretically by the generalized energy-based fragmentation approach under periodic boundary conditions (denoted as PBC-GEBF). Furthermore, different nucleoside molecular morphology (monomer, polymer, and crystal), solvent (implicit and explicit water), and temperature/theoretical model effect on the THz spectra were mainly investigated. It was found that in the low-frequency band, the vibrational modes were generally originated from the collective vibration of all molecules involved (more than 99% of them were vibration; only less than 1% of them were rotation and translation), which can reflect the molecular structure and spatial distribution of different substances. The Gibbs free energy of thymidine monomer, dimer, tetramer, and crystal was studied. It was found that the cell-stacking energy had the greatest influence on the spectrum, indicating that only the crystal structure constrained by the periodic boundary conditions could well describe the experimental results. In addition, hydrophobic forces dominated the formation of new chemical bonds and strong inter-molecular interactions; the free water had little contribution to the THz spectrum of nucleosides, while crystalline water had a great influence on the spectrum.

## 1. Introduction

Nucleosides are the major components of nucleic acids, DNA and RNA, composed of nitrogen-containing bases and ribose or deoxyribose, and play an important physiological role in human immunity, metabolism, liver, cardiovascular, and nervous system, and have antibacterial and antiviral activities [1]. As a fast and effective nondestructive testing method, THz spectroscopy has been applied to crystal engineering, polymer materials, pharmacy, biochemistry, life science, and many other fields [2,3]. Recently, an increasing number of researchers have found that in the THz band (0.1–10 THz), the spectral characteristics were caused by the weakest bond or non-bond interaction of low-frequency molecular internal vibration, such as van der Waals forces, dispersion forces, and hydrogen bonds [4,5]. Water is a polar molecule, and when the THz wave passed through it, it would form an interaction between the substance and water in the picosecond time domain; resonance and relaxation phenomena appeared in the far-infrared and microwave bands, resulting in the powerful absorption of THz waves and drowning out the useful signal [6]. We have reported the THz spectra of DNA/RNA bases, base pairs, and characteristic fragments in our previous work [7,8,9,10,11,12,13,14]. Due to instrument and equipment limitations, all the research was limited to solid samples and a vacuum-testing environment. However, many biological samples can only maintain their unique molecular configuration (such as A/B/Z, the three common configurations of DNA) in an aqueous solution. The physical and chemical properties of an anhydrous substance and a hydrate were quite different. These differences would lead to changes in their stability and bioavailability [15]. It was clear that when water was used as a solvent to form pseudopolymorphs with molecules, the strong absorption brought by it was difficult to remove. However, water molecules reacted with substances to form a eutectic, or connected with solid samples by hydrogen bonds in the way of binding water. How does the water affect the THz spectrum? In addition to the solvation effect, what effects do different initial structures, temperatures, and calculation models have on the THz spectra?

In this paper, we studied the THz spectra of five nucleosides in the solid and liquid phases experimentally by Fourier-transform infrared spectroscopy (FTIR) in a wide spectral region ranging 0.5–9 THz. In addition, we analyzed the lattice energy, geometric structure, and vibration spectrum of molecular crystal theoretically by the generalized energy-based fragmentation approach under periodic boundary conditions (denoted as PBC-GEBF), and obtained the vibrational modes corresponding to all characteristic frequencies by density functional theory (DFT). Different temperatures (0 K and 298 K), initial structures (monomer, polymer, and crystal), solvation effects (gas and liquid phases), and calculation models (harmonic and anharmonic oscillator models) were considered, which was helpful for analyzing the physiological function of the nucleoside and its derivatives.

## 2. Results and Discussion

Our group has conducted meticulous research on the THz spectra of bases, base pairs, and DNA characteristic fragments, and analyzed their spectra and corresponding vibrational modes [7,8,9,10,11,12,13,14]. However, all the experimental samples were based on the solid state, and the theoretical calculation was completed in the gas phase and room temperature environment, without considering the influence of temperature and solvent under the actual environment. In this paper, Fourier-transform infrared spectroscopy (FTIR), combined with the theories of PBC-GEBF and DFT, was used to study the THz broad spectrum of nucleosides. The spectra of nucleoside crystals, dimers, and monomers under different temperature and solvent environments were learned, and the effect of the temperature and the mechanism of water action on the spectrum were finally revealed.

### 2.1. The Morphology Affects the Calculation Results

Figure 1 demonstrated the calculation (single molecular) and experimental spectra of five nucleosides ranging from 0 to 9 THz using a 6 μm-thick Mylar. Figure 1 showed that the calculation and experimental spectra were not in good agreement; in the low-frequency band below 4 THz, especially, the weak interactions between molecules played a major role. As the frequency moved higher, the intra-molecular interactions became stronger with the weakening of the inter-molecular interactions, making the calculation and experimental spectra more consistent.

After comparing the theoretical spectra of bases [9] and nucleoside monomer structure, it was found that the nucleosides showed more abundant characteristic peaks in the THz band, because pentose groups made them non-planar structure, and thus there were more intra-molecular weak interactions (hydrogen bonds), while bases were almost planar structures, and thus there was almost no intra-molecular weak interaction, only intra-molecular tension.

Figure 2 demonstrated the structures of thymidine dimer and tetramer, and the calculated spectra of them. The polymer showed more abundant vibrational modes by comparing with the monomer one in Figure 1b, but it should be noted here that neither monomer, dimer, or polymer can describe the vibrational modes of the crystal considering boundary conditions in the THz band, which was quite different from the experimental spectrum.

The Gibbs free energy of thymidine monomer, dimer, tetramer, and crystal was calculated. The energy of the monomer was −875.191278 hartree, and the energy of dimer, tetramer, and crystal were −1750.375531, −3500.763052, and −3498.811113 hartree, respectively. The interaction between molecules can be expressed by the following formula:$$\Delta E_{{}^{int}}=E_{{}^{T}}-{\sum E}_{i}+E_{BSSE}$$
where Δ*E_int_* was the inter-molecular interaction energy, *E_T_* was the total energy, and ∑*E_i_* was the sum of the energy of each molecular system. *E_BSSE_* was the energy of the basis set superposition error (BSSE), whose effect decreased with the improvement of the basis set. The B3LYP/6-311++G (d,p) basis set considered the dispersion function selected in our paper, so its effect could be ignored here. Therefore, we could calculate that the inter-molecular interaction energy from the thymidine monomer to dimer was about 4.4 kCal/Mol (−1750.375531−2*(−875.191278) = 0.007026 hartree, 1 hartree = 627.5 kCal/Mol). From the dimer to tetramer, it was about −7.5 kCal/Mol. The inter-molecular interaction energy from the tetramer to crystal was about 1224.8 kCal/Mol. Without considering the influence of *E_BSSE_*, the inter-molecular interaction energy presented a small negative value from the dimer to tetramer, but it was not difficult to find that the inter-molecular interaction energy increased significantly from the polymer to crystal, which indicated that the accumulation energy of lattices contributed greatly, so it was necessary to consider periodic boundary conditions.

Therefore, in order to make the calculation results more consistent with the experimental ones, the initial structure must be based on the crystal. The weak interaction between molecules and the stacking force of the lattice should be considered in the theoretical calculation. Figure 3 showed the single-cell structure of thymidine with hydrogen bonds and Van der Waals forces, and the comparison of the experimental and calculation (gas phase and crystal structure) spectra of it. The corresponding absorption peaks of both the calculation and experimental spectra were marked. It can be seen from Figure 3 that the PBC-GEBF method with periodic boundary conditions can fully describe the experimental environment.

Table 1 displayed all the absorption peaks and their corresponding vibrational modes of thymidine. In comparison with the calculation spectrum, except for the degeneracy of some weak absorption peaks in the experimental spectrum due to the insufficient accuracy of the test equipment, the rest were consistent. The potential energy distribution (PED) method can quantitatively describe the contribution of a given group of atoms in each vibrational mode, and with it we can obtain the contribution percentage of each vibration. Therefore, it can also be seen from the table that in the low-frequency band below 6 THz, the vibrational modes mainly came from the collective vibration participated by all molecules (more than 99% of them were vibration; only less than 1% of them were rotation and translation), and the out-plane distortion contributed a lot, accounting for more than 90%. When the frequency went up, the vibrational modes changed into the out-plane and in-plane bending and swing involving some molecules and atoms.

### 2.2. Temperature and Calculation Models Affect the Results

In the THz band, the experimental spectrum shifted due to the influence of the test environment temperature, but during the theoretical simulation, the temperature had no effect on the spectrum. It was because quantum mechanics used the harmonic oscillator model to solve the Schrodinger equation to calculate the vibration frequency without taking into account temperature. Figure 4a calculated the vibrational spectra of cytidine monomer in the gas phase by using harmonic and anharmonic oscillator models with the default temperature (298.15 K). There was no difference between them. Figure 4b exhibited the spectrum calculated by an anharmonic oscillator model at different temperatures. The results were in good agreement with each other. Figure 4c was the spectra of cytidine monomer in the implicit solvent (water) environment at 0 K and 298 K, respectively. They showed a high degree of consistency. However, with the addition of the water solvent, the spectrum showed a slight difference, which can be seen from Figure 4d. Comparing Figure 4a,b, we could find that the selection of temperature and calculation model in the gas phase environment would not affect the calculation results. However, after the addition of the water solvent, the spectrum based on different calculation models changed obviously, mainly characterized by the change of peak intensity and the blue shift of the frequency, which can be discovered by a comparison of Figure 4a,c,d. That was because the solvent was equivalent to a continuous electrolyte, which would smooth the potential energy surface, and solvation could make a greater impact on the vibration of the larger dipole moment.

### 2.3. Solvation Affects the Calculation Results

For liquid samples, different concentrations and pH values could change the sample properties (such as DNA configuration), and the polarization orientation and thickness of samples could also bring some differences to the results [16]. Therefore, it was necessary to find the optimal test scheme through repeated experiments. In addition, when bio-molecules were in the water solvent, some interactions (such as hydrogen bonds) were formed between them, which could submerge the effective spectral characteristics of the sample itself. Therefore, accurately extracting the effective information of complex biological system under water interference, studying the mechanism of the interaction between macromolecules and water, then reflecting it on the spectral changes was of great significance to understanding the biological differences of various nucleoside derivatives and the conformational transformation mechanism of DNA/RNA. Moreover, many substances exist in nature in the form of crystalline hydrate, and the influence of crystalline water cannot be ignored.

In order to explore the influence of the water solvent, the THz spectra of five nucleoside liquid samples (pseudopolymorphs) were measured. The 25 μm-thick Mylar was selected, the effective spectral range was 0.5–4 THz, and the thickness of the sample in the liquid sample pool was about 100 μm. In Figure 5, pure water with the same thickness of samples was selected as the background, while the air was selected as the test background in Figure 6. The optical cavity was vacuumed, and the sample chamber was at atmospheric pressure under room temperature. We chose different test backgrounds (water and air) to try to find the influence of the solvent water on the spectrum.

Figure 5 showed that the THz spectra of solid and liquid samples of adenosine and thymidine had good repeatability after several measurements (the spectra were completely original spectra without any treatment), but the coincidence was not perfect due to the influence of the solvent water. Figure 6 showed the comparison of the THz spectra of cytidine, guanosine, and uridine in the solid and liquid phases with air as the background. The original spectrum (the average of three test results) and the filtered spectrum were shown, respectively. Similarly, the solid and liquid spectra were roughly consistent. The liquid and solid spectral characteristics of guanosine were weaker than those of cytidine and uridine. However, due to the influence of the water solvent in the liquid sample, the position of some characteristic peaks also appeared—frequency shift and mismatch. The positions of characteristic peaks in the THz spectra of all nucleosides in the solid and liquid samples were given in Table 2.

Table 2 displayed the free water as the solvent around the molecules, which may have had a minor impact on the THz spectrum. However, as a polar molecule, the absorption of water in the THz band was strong. Even if we tried to remove the influence of water through background subtraction, the general method could not achieve an accurate deduction.

In addition to free water, the effect of crystal water on the THz spectrum could not be ignored. Figure 7 showed the experimental and calculated spectra of guanosine. In the experiment part, the guanosine anhydrous crystal was selected as the test sample, while the guanosine dihydrate from CSD (as shown in Figure 7a) was selected as the initial structure in the theoretical calculation. It can be seen from the Figure 7 that there was some inconsistency between the theoretical and experimental spectra especially between 3–6 THz, which was speculated to be related to the crystal water in the initial structure. Table 3 displayed the positions of all characteristic peaks and corresponding vibrational modes in the theoretical and experimental spectra.

Table 3 displayed the positions of all characteristic peaks and corresponding vibrational modes in the theoretical and experimental spectra for anhydrous guanosine powder and dihydrate guanosine crystal. We could find that water molecules were involved in all vibrational modes in the THz band, and showed very strong vibration and activity at 3.69 and 4.77 THz, forming rich hydrogen bonds with guanosine molecules.

Next, let us carefully explore the effect of crystal water and free water on the THz spectrum. Figure 8a compared the calculation spectra of cytidine monomer in the implicit water solvent (named Cytidine-water, simulated free water state) and with that in thet gas phase (named Cytidine) environment. The vibrational modes of the Cytidine-water near 6.2 and the Cytidine near 6.5 THz both came from the out-of-plane distortion of ribose. The vibrational modes of the Cytidine-water at 7.3 THz and the Cytidine at 7.1 THz both originated from the weak out-of-plane distortion of ribose and out-of-plane swing of NH_2_ attached to the pyrimidine ring. The vibrational modes of the Cytidine-water at 8.6 THz came from the out-of-plane distortion of O-H attached to ribose, while the vibrational modes of the Cytidine at 9.0 THz came from the out-of-plane swing of NH_2_ attached to the pyrimidine ring. It was found in the THz band, although free water molecules formed rich hydrogen bonds around the structure, this only led to the red shift of the spectrum; the overall envelope of the spectrum, the absorption peaks, and the mainly vibrational modes did not change significantly.

Figure 8b showed the THz spectra of the guanosine dimer in the gas phase (named Guanosine-dimer), the implicit water solvent (named Guanosine-dimer-water, simulated free water state), and the explicit water solvent (named Guanosine-dimer-2H_2_O, simulated bound water state. Appendix A showed the initial structure of Guanosine-dimer-2H_2_O (the CSD reference code is GUANSH10). The initial structures of Guanosine-dimer and Guanosine-dimer-water were the structure of Appendix A by removing the water molecules, and adding the implicit solvent water around after removing the water molecules separately. The optimized structures of Guanosine-dimer-2H_2_O, Guanosine-dimer, and Guanosine-dimer-water were displayed in Appendix A–d. It can be found that the optimized structure of Guanosine-dimer-2H_2_O was excessively different from the initial one due to the strong hydrogen bonds between the molecules and water. In the optimized structure, the guanosine and water molecules were more compact. The calculation results showed that they all had three absorption peaks in the range of 0–10 THz. Appendix A showed the vibrational modes corresponding to the main characteristic peaks of Guanosine-dimer-2H2O, Guanosine-dimer, and Guanosine-dimer-water. The vibrational modes of the Guanosine-dimer near 5.5 THz, and the Guanosine-dimer-water near 6.2 both originated from the out-of-plane swing of O-H on ribose, while the vibrational modes of the Guanosine-dimer near 8.7 THz, and the Guanosine-dimer-water near 9.5 THz both came from the out-of-plane swing of NH_2_ attached to the purine ring. The vibrational modes of the Guanosine-dimer near 7.1 THz, and the Guanosine-dimer-water near 7.2 THz both came from the out-of-plane swing of NH_2_ attached to the purine ring accompanied by thet out-of-plane swing of O-H on ribose. It could be seen that the implicit aqueous solvent has little effect on the characteristic peaks and vibrational modes, which may be due to the water molecules and other molecules exceeding the bonding distance of the hydrogen bonds. However, the crystalline water was different, as can be seen from Appendix A; the vibrational modes of the Guanosine-dimer-2H_2_O near 5.7 THz mainly came from the out-of-plane swing of water molecules, and the vibrational modes near 9.7 THz also include water molecules.

## 3. Procedure for Experiment

The nucleoside samples (adenosine, thymidine, cytidine, guanosine, and uridine) used in the experiment are white or near-white powder with a purity of more than 99%, and thymidine is deoxynucleoside. All the samples were purchased at Sigma-Aldrich Chemical Co. (St. Louis, MO, USA), without any purification treatment before testing. Figure 9 showed the optimized monomer structures of nucleoside.

In the experiment, solid and liquid samples were selected for measurement. Before testing, each solid sample was mixed with a certain proportion of polyethylene powder (PET; the mass ratio of the sample to polyethylene is generally between 1:3–1:5, depending on the absorption strength of the sample in THz band). After even grind, these solid samples were pressed into a thin sheet with a diameter of 13 mm and a thickness of about 1 mm. To eliminate the influence of reference, the weight of PET in the background should be consistent with that in the sample. Each liquid sample was diluted with pure water, making them saturate and dissolve, and then placed on a vortex shaker for 3 min. After being kept at room temperature for 10 min, the upper transparent liquid was injected into the experimental liquid vessel for testing. The thickness of the sample was about 100 μm, and the same thickness of pure water was regarded as the background in order to ignore the effect of water.

THz spectra were measured by Bruker v80v FTIR with the frequency ranging from 33 to 680 cm^−1^ (0.9–20.4 THz) with 6 μm-thick Mylar and 10 to 150 cm^−1^ (0.3–4.5 THz) with 25 μm-thick Mylar under room temperature. The sample cavity and optical cavity were both vacuumed during the solid sample testing; in contrast the optical cavity was the sole aspect vacuumed during the liquid sample testing, and the former was maintained at atmospheric pressure. The principles of FTIR were introduced in a previous study [8] and were skipped in this paper. The spectra of samples had a good consistency after several runs.

## 4. Theoretical Method

The crystal and polymer structures used for the theoretical calculation were all from the Cambridge Structural Database (CSD). Their CSD reference codes were: THYDIN (thymidine crystal, displayed in Figure 3) [17] and GUANSH10 (guanosine dihydrate, displayed in Figure 7) [18].

For crystals, efforts have been made to develop linear-scaling quantum chemical algorithms that can deal with macromolecules and periodic structure. The commonly used calculation methods were Hartree–Fock or DFT under periodic conditions, and the popular software used include Crystal14, Materials Studio, VASP, etc. However, for crystal structures with more molecules, there are some problems such as a high computational cost and difficult convergence for these traditional quantum chemistry calculation methods. To do this, we introduced the PBC-GEBF (the generalized energy-based fragmentation approach under periodic boundary conditions) method, which expresses the total ground state energy of a macromolecular system as a linear combination of the ground state energy of a series of smaller subsystems or of “electrostatically embedded” subsystems. Using this method, we can perform full quantum mechanical calculations on a common server for molecular systems containing thousands of atoms to obtain accurate lattice energy, geometric structure, and vibrational spectrum. The improved PBC-GEBF method [19,20,21,22] under Gaussian09 software was first adopted to calculate the vibrational spectrum of thymidine and guanosine dihydrate crystal after the structure was fully optimized without any virtual frequency. By converting the calculation of energy and other properties of molecular crystals into the calculation of a series of small molecular clusters, the computational cost was significantly reduced under the condition of maintaining the same accuracy. The basic principle started with the division of the molecular structure into several independent segments, and then a series of subsystems containing several molecules were generated systematically from the molecular slices. Secondly, each subsystem was embedded into the electrostatic field composed of a group of background point charges, and a compensation field charge was added at the boundary to consider the polarization and long-range electrostatic interaction of the real infinite periodic crystal charge background on the subsystem. In this method, the energy of a single cell can be obtained by combining the energies of a series of subsystems according to a certain coefficient. In the multimer expansion of cell energy, all the short-range interactions between molecules were required to be calculated only once. Considering the dispersion effect, the PBE(d3bj)/6-311++G(d,p) base set was selected for the crystal calculation with the full width at half maximum (FWHM) being 4 cm^−1^.

For monomer and polymer structures, we only considered the intra-molecular interaction, using the B3LYP/6-311++G(d,p) method under Gaussian16 software to fully relax the structures and to calculate the vibration frequencies and corresponding vibrational modes. All computation results were rendered by using Mercury and GaussView 6.0, respectively.

## 5. Conclusions

How to accurately extract the component information of a complex biological system under the interference of water, and accurately grasp the mechanism of the water action (solute–solute, solvent–solvent, or solute–solvent intermolecular interaction) had become an obstacle in the THz research field. In this paper, the THz spectra of five nucleosides in the solid and liquid state were studied by the Fourier-transform infrared spectroscopy (FTIR) combined with DFT and the PBC-GEBF methods. The PBC-GEBF method, which took into account the intra- and inter-molecular interactions as well as the periodic boundary conditions of molecules, can well describe the THz vibrational spectra of solid molecules with a small computational cost. In the low-frequency band (generally lower than 6 THz), the vibrational modes generally originated from the collective vibration of all molecules in the lattice; out-plane and in-plane torsional and stretching vibrations contribute greatly, about more than 90%. With the increase of the frequency, the vibrational modes changed into the vibration of some molecules or some atoms, which was consistent with our previous research results. Neither a single molecule nor polymer can describe the THz spectrum of the sample, which solely made the results meaningless.

The calculation spectra of the gas-phase solid structure on the basis of the harmonic oscillator and the anharmonic oscillator model had no difference, but when the implicit solvent was added around the structure, the calculation spectra of the two models would be different. That was because the solvent was equivalent to a continuous electrolyte, and would smooth the potential energy surface. A low temperature could improve the detection accuracy and broaden the frequency detection range. However, for the theoretical calculation, the temperature had no effect on the spectrum. By comparing the experimental spectra of the solid with the liquid nucleoside samples, it was indicated that even if the influence of water (free water) was removed by data-processing methods, there was still a difference between them. As a solvent, free water could form rich hydrogen bond interactions with biomolecules, but it had a minor impact on the spectrum because of the water molecules and other molecules exceeding the bonding distance of the hydrogen bonds. Crystal water was different and had a great influence on the THz spectrum. The research provided a theoretical basis for simulating the actual biological environment, developing new nucleoside drugs, and analyzing protein and DNA/RNA.

## Figures and Tables

**Figure 1 molecules-28-01528-f001:**
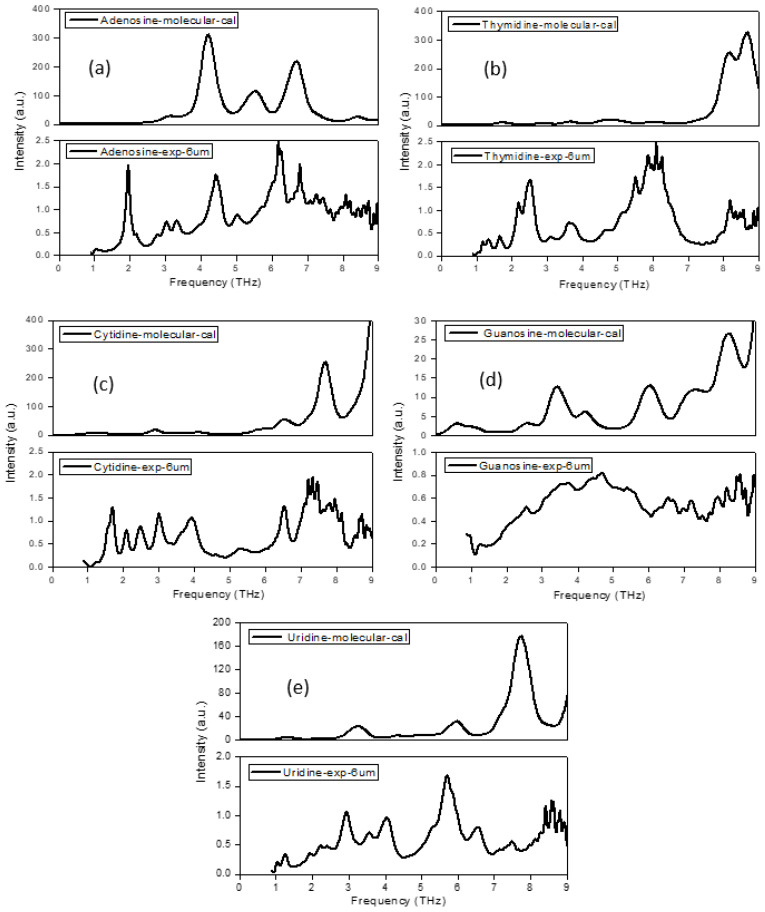
Comparison of calculation (single molecular) and experimental spectra of five nucleosides.

**Figure 2 molecules-28-01528-f002:**
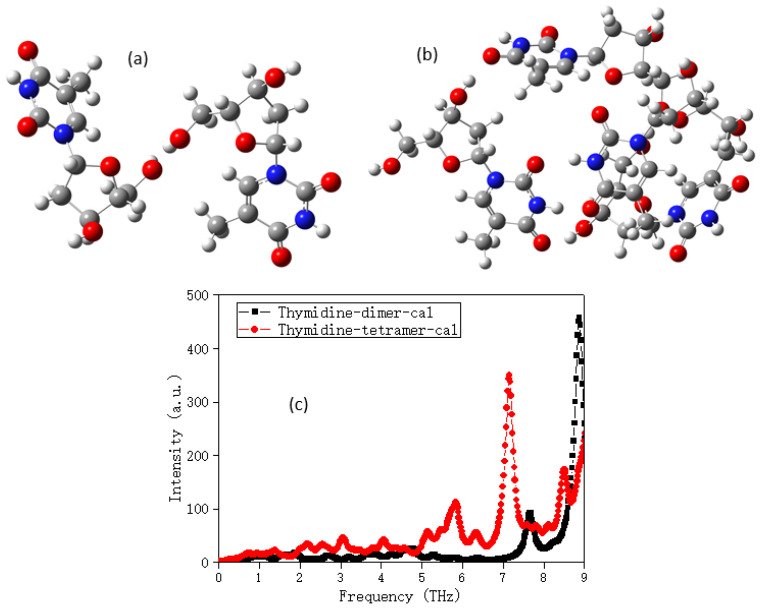
The structures of thymidine (**a**) dimer and (**b**) tetramer; (**c**) the calculated spectra of them.

**Figure 3 molecules-28-01528-f003:**
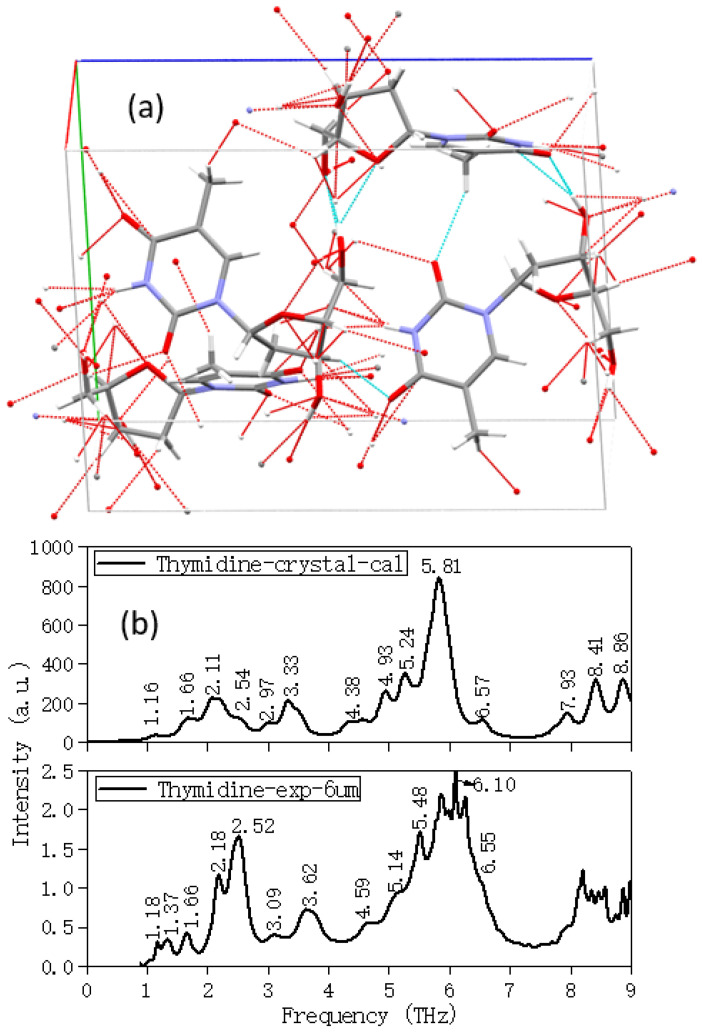
(**a**) The single-cell structure of thymidine with hydrogen bonds and Van der Waals forces, (**b**) comparison of experimental and calculation (crystal) spectra of thymidine based on PBC-GEBF method by using Gaussian09 software.

**Figure 4 molecules-28-01528-f004:**
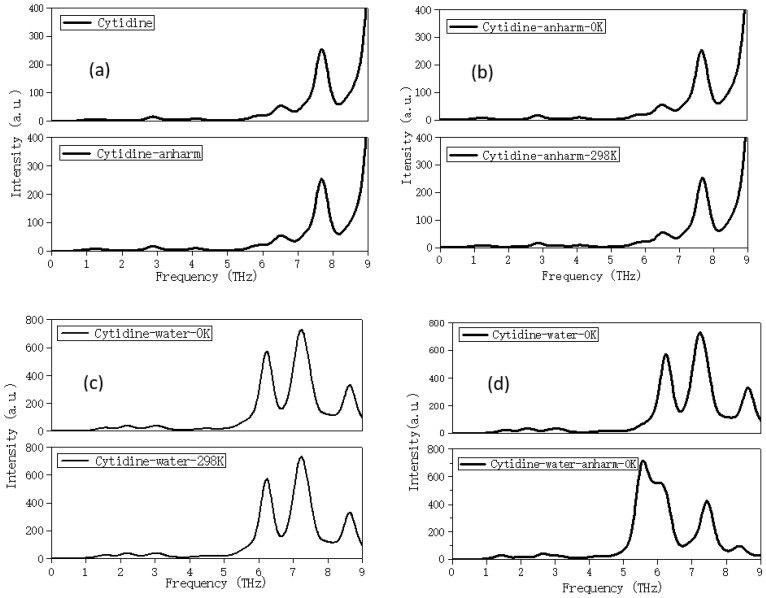
Comparison of the effects of (**a**) different models without solvent, (**b**) different temperatures without solvent, (**c**) different temperatures with solvent, and (**d**) different models with solvent, on the calculation spectra.

**Figure 5 molecules-28-01528-f005:**
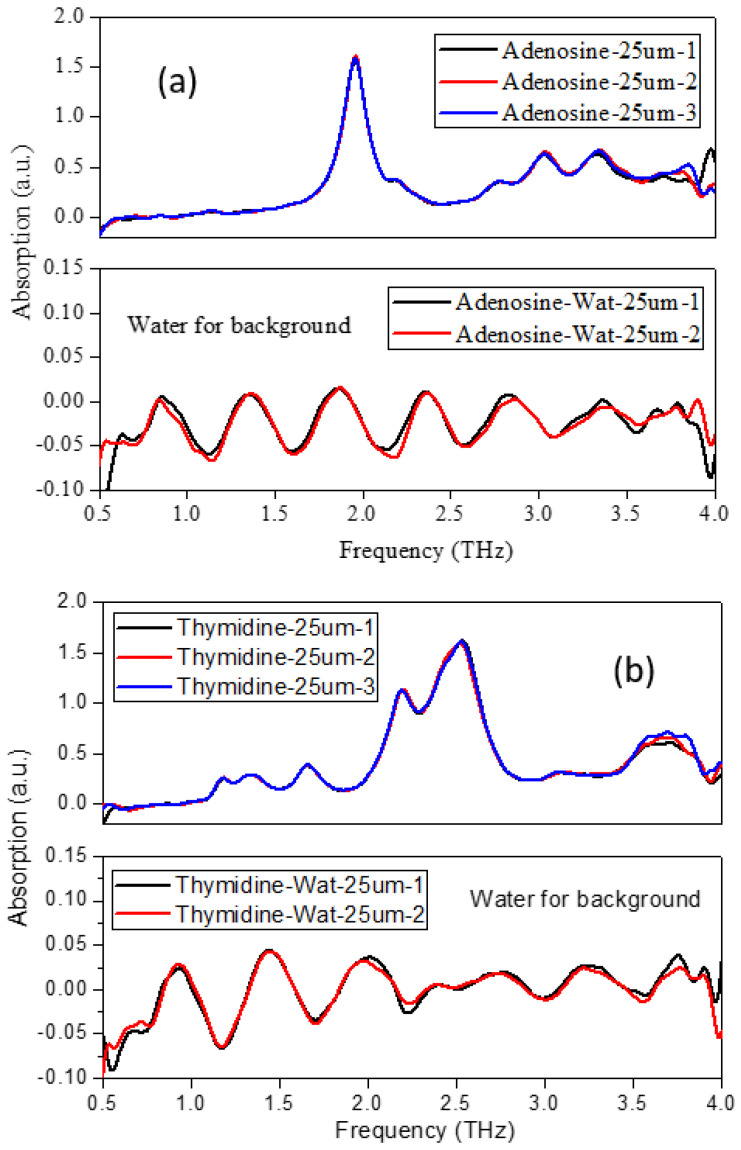
Comparison of THz spectra of solid and liquid samples—(**a**) adenosine, (**b**) thymidine—with water for background.

**Figure 6 molecules-28-01528-f006:**
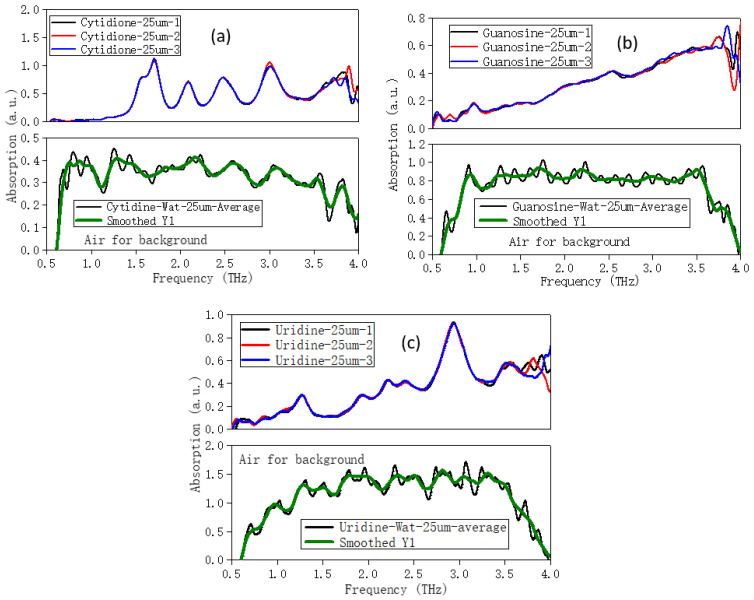
Comparison of THz spectra of solid and liquid samples—(**a**) cytidine, (**b**) guanosine, (**c**) uridine—with air for background.

**Figure 7 molecules-28-01528-f007:**
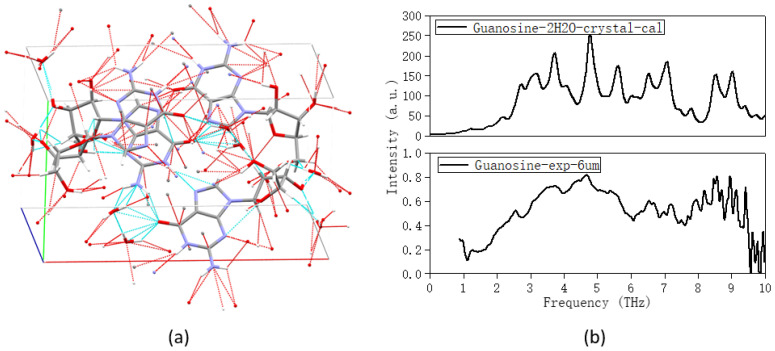
(**a**) The crystal structure of guanosine dihydrate; (**b**) comparison of the calculated (guanosine dihydrate, crystal structure) and experimental (crystalline guanosine) spectra.

**Figure 8 molecules-28-01528-f008:**
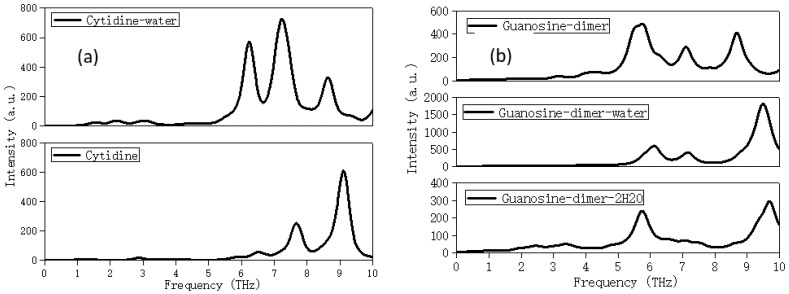
(**a**) The calculated spectra of cytidine single molecule using harmonic oscillator models, the initial structure in the implicit water solvent and gas phase, respectively. (**b**) The calculated spectra of guanosine dimer in the gas phase, with implicit and explicit solvent, respectively.

**Figure 9 molecules-28-01528-f009:**
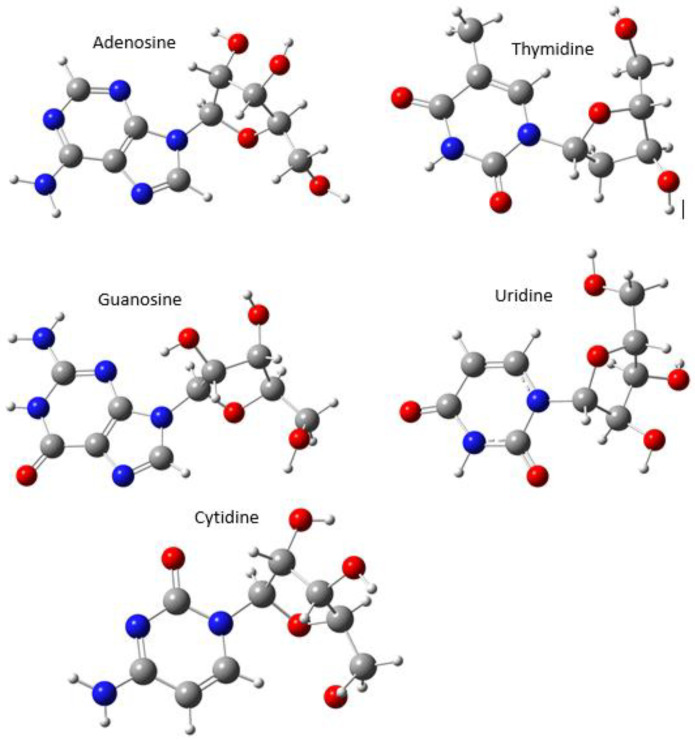
Optimized monomer structures of nucleoside.

**Table 1 molecules-28-01528-t001:** Experimental and calculated THz vibrational frequencies and corresponding vibrational modes for thymidine.

Thymidine			
Experimental		Calculated	
Peaks (THz)		Main Vibrational Modes	Description
1.18	1.16	Collective vibration	Overall out-of-plane swing
1.37			
1.66	1.66	Collective vibration	Overall out-of-plane bending
2.18	2.11	Collective vibration	Overall out-of-plane swing and bending
2.52	2.54	Collective vibration	Overall out-of-plane swing and bending
3.09	2.97	Collective vibration	Overall out-of-plane swing
3.62	3.33	Collective vibration	Overall out-of-plane swing
4.59	4.38	Collective vibration	Out-of-plane swing and bending
5.14	4.93	Collective vibration	Out-of-plane swing and bending
5.48	5.24	Collective vibration	Out-of-plane swing and bending
6.10	5.81	Partial vibration	Partial out-of-plane bending
6.55	6.57	Partial vibration	Pentose bases out-of-plane swing and bending
	7.93	Partial vibration	Purine ring expanding, pentose bases out-of-plane swing
	8.41	Partial vibration	Partial weak out-of-plane bending
	8.86	Partial vibration	CH_3_ on pentose group in-plane swing

**Table 2 molecules-28-01528-t002:** Vibration frequencies (in THz) of five nucleoside solid and liquid samples.

Adenosine		Thymidine		Cytidine		Guanosine		Uridine	
Solid	Liquid	Solid	Liquid	Solid	Liquid	Solid	Liquid	Solid	Liquid
	0.85	1.18	0.93		0.86	0.98	0.93	1.27	1.27
	1.36	1.35	1.45	1.56	1.28	2.05 (weak)	1.71 (weak)		1.49
1.95	1.87	1.66	1.98	1.67	1.47	2.55 (weak)	2.2 (weak)	1.92	1.88
2.75	2.37	2.19	2.38	2.11	2.17	3.74	3.51	2.21	2.31
3.04	2.82	2.52	2.76	2.46	2.59			2.42	2.49
3.35	3.41	3.10	3.26	3.01	3.06			2.93	2.82
3.82	3.80	3.67	3.75	3.93	3.53				3.05
					3.79			3.55	3.32

**Table 3 molecules-28-01528-t003:** Experimental and calculated THz vibrational frequencies and corresponding vibrational modes for anhydrous guanosine powder and dihydrate guanosine crystal.

Guanosine		Guanosine Dihydrate	
Experimental		Calculated	
Peaks (THz)		Main Vibrational Modes	Description
2.57	2.19	Collective vibration	Overall out-of-plane distortion, H_2_O include
3.65	2.69	Collective vibration	Overall in-plane distortion, H_2_O include
	3.19	Collective vibration	Overall out-of-plane swing, H_2_O include
	3.69	Collective vibration	Weak in-plane swing, **H_2_O strong distortion**
4.65	4.77	Collective vibration	Weak out-of-plane distortion, **H_2_O strong distortion**
5.38	5.56	Collective vibration	Out-of-plane distortion, H_2_O weak swing
6.53	6.49	Partial vibration	Out-of-plane distortion of pentose groups, H_2_O weak distortion
7.17	7.05	Partial vibration	Out-of-plane distortion of purine ring, H_2_O weak distortion
	7.82	Partial vibration	**H_2_O strong distortion**
	8.52	Partial vibration	Out-of-plane distortion of pentose groups, H_2_O weak distortion
	9.05	Partial vibration	Out-of-plane distortion of pentose groups, H_2_O weak distortion

## Data Availability

Not applicable.

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
