# Peer review of "Effects of Morphology and Solvent/Temperature on THz Spectra: Take Nucleosides as Example"

_molecules, 2023, doi:10.3390/molecules28041528_

Round 1
Reviewer 1 Report
This paper presents the investigation on the THz spectra of five nucleosides in solid and liquid state combined with calculation. The paper can be accepted on Molecules after the following revisions.
1. There are many grammatical errors, typos, and format errors throughout the manuscript and Figures, please check. For example, there are too many disorderly atoms in Figure 4a and 8a, and some of the serial numbers of illustration are lost or block the pictures.
2. Generally, it would be different in crystal structure between guanosine and guanosine dihydrate, and corresponding interactions would also be different. Since crystal water has significant influence on the THz spectrum, and the authors performed the calculation of guanosine dihydrate, why not verify the experimental spectrum of guanosine dihydrate?
3. In the manuscript, crystal structures were used as model. How the authors consider the experimental THz spectra differences between polycrystalline powder and single crystal plate? They share the same microscopic structure.
4. How to evaluate the consistency between calculation results based on the modified crystal structure model and experimental liquid sample, and whether this method is universal?
5. It is suggested that rewritten the conclusion part to more condensed paragraphs.
Author Response
Dear Reviewer 1:
Thank you very much for your meticulous, professional and all good suggestions. The following describes our responses and revisions based upon your comments.

Reviewer 2 Report
The manuscript describes properties of terahertz spectra of different nucleosides in solid and liquid phases, especially focusing on the weak interactions. The approach combines experimental and computational studies to simulate the effects of morphology, solvent, and temperature on the spectra. In my opinion, the comparison between different kinds of computational approaches with the acquired experimental spectra gives valuable information on the suitability of the adopted models for simulating the real spectra, and therefore I can recommend publication in Molecules. I have only small comments which will be easy to address before publication:
The manuscript could benefit from careful language revision. At least, please correct the computational program name throughout the text (Gaussian, not Guassian). Also, reporting calculated interaction energies with such a great accuracy is not acceptable, for example the interaction of two dimers forming a tetramer (-7.523725 kcal/mol) is really too accurate and I would change it to mere -7.5 kcal/mol. Furthermore, it is not quite clear to me, if the interaction energies included BSSE or not (and how large would it be - in weak interactions BSSE can be larger than the actual interaction energy). Please clarify.
Author Response
Dear Reviewer 2:
Thank you very much for your meticulous, professional and all good suggestions. The following describes our responses and revisions based upon your comments.

Round 2
Reviewer 1 Report
This work is acceptable in this version.